# Transcriptome Profiling Reveals the Gene Network Responding to Low Nitrogen Stress in Wheat

**DOI:** 10.3390/plants13030371

**Published:** 2024-01-26

**Authors:** Yiwei Wang, Pengfeng Li, Yiwang Zhu, Yuping Shang, Zhiqiang Wu, Yongfu Tao, Hongru Wang, Dongxi Li, Cuijun Zhang

**Affiliations:** 1College of Computer Science and Technology, Taiyuan University of Technology, Taiyuan 030024, China; wangyiwei0703@163.com; 2Shenzhen Branch, Guangdong Laboratory of Lingnan Modern Agriculture, Key Laboratory of Synthetic Biology, Ministry of Agriculture and Rural Affairs, Agricultural Genomics Institute at Shenzhen, Chinese Academy of Agricultural Sciences, Shenzhen 518120, China; pengfengli17@126.com (P.L.); zhuyiwang@caas.cn (Y.Z.); shangyuping168@163.com (Y.S.); wuzhiqiang@caas.cn (Z.W.); taoyongfu@caas.cn (Y.T.); wanghongru@caas.cn (H.W.); 3College of Life Sciences, South China Agricultural University, Guangzhou 510642, China; 4College of Agronomy, Shanxi Agricultural University, Jinzhong 030801, China

**Keywords:** wheat, low nitrogen, RNA-seq, nitrogen signaling network

## Abstract

As one of the essential nutrients for plants, nitrogen (N) has a major impact on the yield and quality of wheat worldwide. Due to chemical fertilizer pollution, it has become increasingly important to improve crop yield by increasing N use efficiency (NUE). Therefore, understanding the response mechanisms to low N (LN) stress is essential for the regulation of NUE in wheat. In this study, LN stress significantly accelerated wheat root growth, but inhibited shoot growth. Further transcriptome analysis showed that 8468 differentially expressed genes (DEGs) responded to LN stress. The roots and shoots displayed opposite response patterns, of which the majority of DEGs in roots were up-regulated (66.15%; 2955/4467), but the majority of DEGs in shoots were down-regulated (71.62%; 3274/4565). GO and KEGG analyses showed that nitrate reductase activity, nitrate assimilation, and N metabolism were significantly enriched in both the roots and shoots. Transcription factor (TF) and protein kinase analysis showed that genes such as MYB-related (38/38 genes) may function in a tissue-specific manner to respond to LN stress. Moreover, 20 out of 107 N signaling homologous genes were differentially expressed in wheat. A total of 47 transcriptome datasets were used for weighted gene co-expression network analysis (17,840 genes), and five TFs were identified as the potential hub regulatory genes involved in the response to LN stress in wheat. Our findings provide insight into the functional mechanisms in response to LN stress and five candidate regulatory genes in wheat. These results will provide a basis for further research on promoting NUE in wheat.

## 1. Introduction

Wheat (*Triticum aestivum*; 2n = 6x = 42; AABBDD) is the third most important crop globally, feeding ~30% of the world’s population [1]. Wheat yields will need to be constantly increased to cope with the increasing pressure of population growth until 2050 [2]. Nitrogen (N) is an essential nutrient for plant growth and development and is also a component of cellular molecules, such as amino acids, nucleic acids, and chlorophyll [3,4]. The application of N fertilizer has substantially increased wheat yields in the past few decades. Given the low N use efficiency (NUE) of wheat (less than 40%), over-fertilization has also caused serious environmental pollution [5,6,7,8]. Therefore, understanding the mechanism of NUE and improving NUE in wheat are of paramount importance.

Nowadays, many N-related genes have been identified in plants, including structural genes, transcription factors (TFs), and protein kinases (PKs) [9,10]. Among these genes, TFs and PKs constitute the N signaling pathway in response to low N (LN) stress [9,10,11,12,13]. In general, the signaling pathway had three types of genes, including N signaling perception genes, regulatory hub genes, and other regulatory genes (ORGs). In the model plant *Arabidopsis*, AtNPF6.3 (AtNRT1.1) is a dual-affinity nitrate transporter, also known as an external N level sensor [14,15,16,17]. N signaling is converted to calcium signaling via AtNPF6.3 at the N signaling perception stage [14,15]. Intracellular calcium and N signaling are further efficiently tuned by AtCNGC15 [18]. As calcium concentrations increase, AtCPK10/30/32 phosphorylates AtNLP6/7 and promotes the nuclear retention of AtNLP7 to activate the LN response [16,19]. Then, AtNLP6/7 interacts with several TFs, such as AtNRG2, AtTCP20, and AtHBI1, to regulate the downstream ORGs [20,21,22,23,24]. In addition, these ORGs are further divided into two functional groups in response to LN stress. The first subgroup of genes is positively regulated by LN stress under the *AtNLP6/7* downstream, including *AtTGA1/4*, *AtbZIP1*, *AtNAC56*, and *AtANR1* [25,26,27,28,29,30]. For example, *AtTGA1/4* regulates the expression of *AtNRT2.1* and *AtNRT2.2* in response to N stress [25]. Overexpression of *AtANR1* activates the downstream of *AtNRT1.1* and also promotes root growth under LN stress [30]. The second subgroup of genes is negatively regulated in response to LN stress under the *AtNLP6/7* downstream, including *AtHRS1/HHO1*, *AtLBD37/38/39*, *AtBT1/2*, and *AtSPL9* [31,32,33,34,35,36,37]. For example, *AtHRS1/HHO1* negatively regulates the expression of *AtNRT1.1* and *AtNRT2.4* [31,32,33]. In the *AtBT1/2* double mutant, the expression of *AtNRT2.1* and *AtNRT2.4* is increased [36]. In summary, the identification of these genes has given us a good understanding of the response mechanisms of plants under LN conditions.

With the development of biotechnology, transcriptomic data have become an important tool for understanding the potential mechanisms of gene expression [38,39]. Over the past decades, a large amount of N-related transcriptomic data has been analyzed in crops to understand the N response mechanisms [40,41,42,43,44,45,46,47,48,49,50,51,52,53,54,55,56,57,58]. In rice, the expression of genes related to energy metabolism, amino acid metabolism, and carbohydrate metabolism was reduced under LN stress [40]. In barley, LN tolerance-related genes are mainly associated with amino acid metabolism, starch and sucrose metabolism, and secondary metabolism [41]. A study under LN stress found that the alanine, aspartate, and glutamate metabolism, terpenoid backbone biosynthesis, and vitamin B6 metabolism pathways play a key role in wheat NUE [54]. Genes related to nitrogen compound metabolism, carbon metabolism, and photosynthesis are altered during chronic nitrogen stress in durum wheat [55].

Although several transcriptomic datasets of wheat LN tolerance are available, this research has rarely focused on the signaling transduction mechanism in wheat [51,52,53,54,55,56,57,58]. Therefore, we aimed to understand the LN response and N signaling transduction mechanisms in wheat. In this study, we first obtained the wheat LN stress transcriptome data. Then, we identified genes of the N signaling pathway in wheat and analyzed their expression patterns. Finally, we performed weighted gene co-expression network analysis (WGCNA) using the published data and identified five potential LN response TFs. Our results will help develop understanding of the LN response mechanism in wheat and provide gene resources for further study of N signaling transduction mechanisms.

## 2. Results

### 2.1. Wheat Seedling Morphology under Low Nitrogen Stress

Compared to normal nitrogen (NN) conditions, the phenotypes of the wheat seedlings changed significantly after 12 days of LN conditions (Figure 1A,B). Seedlings under LN stress had shorter shoot lengths and lower fresh and dry weights compared to seedlings under normal conditions (Figure 1C–E). However, seedling root length, fresh weight, and dry weight increased significantly by 17.86% (28 to 33 cm), 41.18% (0.17 to 0.24 g), and 46.15% (0.013 to 0.019 g) under LN conditions, respectively (Figure 1C–E). LN stress resulted in increased wheat root growth, while shoot growth was inhibited, indicating that the growth balance of wheat was significantly affected under LN conditions.

### 2.2. Transcriptomics Quality and Mapping Statistics

To investigate the transcriptome shift induced by LN stress in wheat shoots and roots, a total of 12 cDNA libraries were constructed, including LN root 1 (LR1), LN root 2 (LR2), LN root 3 (LR3), LN shoot 1 (LS1), LN shoot 2 (LS2), LN 3 (LS3), NN root 1 (NR1), NN root 2 (NR2), NN root 3 (NR3), NN shoot 1 (NS1), NN shoot 2 (NS2), and NN shoot 3 (NS3). Due to the poor data quality of NR1, the remaining 11 samples were used for subsequent analysis. In total, 507.60 million clean reads were obtained from these 11 samples. The sequencing error rate was less than 0.03%, the average GC content was 55%, and the percentages of Q20 and Q30 were over 97.32% and 92.35%, respectively. The total percentage of mapped reads ranged from 97.18% to 98.63%, including unique mapped reads, from 87.72% to 91.27% (Table 1). In general, these results indicated the high quality of the data in this study.

Moreover, Pearson’s correlation coefficient and principal component analysis (PCA) were used to estimate the repeatability of the samples. Pearson correlation values ranged from 0.82 to 0.99, indicating qualified biological replicates (Figure 2A). PCA analysis also showed a good correlation between biological replicates (Figure 2B). Based on the PCA clusters, there were different LN response patterns between shoots and roots (Figure 2B). Notably, the root data were further divided into two groups (LN root and NN root) according to the amount of nitrogen applied, suggesting transcriptome divergence. In summary, these high-quality data supported the subsequent in-depth analysis in this study.

### 2.3. Identification and Analysis of Differentially Expressed Genes

Differentially expressed genes (DEGs) reflect internal changes and suggest potential mechanisms for plant responses to environmental and growth changes. In this study, a total of 8468 DEGs (*p*-value < 0.5 and |${log}_{2}$Foldchange| ≥ 1) were identified from the 62,128 detective transcripts. Compared to the NN conditions, the number of DEGs under LN stress was 4467 in the roots and 4565 in the shoots. It is noteworthy that although the number of DEGs in the roots and shoots was similar, their response patterns to LN were distinctly different. Compared to NN conditions, 2955 DEGs were up-regulated and 1512 DEGs were down-regulated in the roots (Figure 3B–E). In the shoots, 1291 DEGs were up-regulated and 3274 DEGs were down-regulated compared to NN conditions (Figure 3A,C,D). In addition, 564 (6.7%) DEGs were differentially expressed in both the roots and shoots, of which 81 DEGs were up-regulated and 89 DEGs were down-regulated (Figure 3F). The remaining 394 DEGs showed opposite expression patterns in the roots and shoots (Figure 3F). Collectively, our results indicated that the response mechanisms of the roots and shoots to LN are different.

### 2.4. GO and KEGG Enrichment of Differentially Expressed Genes

Understanding the potential function of DEGs is helpful to elucidate the molecular mechanisms by which plants respond to environmental stimulation. In this study, given the different response patterns between wheat shoots and roots, Gene Ontology (GO) and Kyoto Encyclopedia of Genes and Genomes (KEGG) analyses were applied to uncover the potential LN response mechanisms.

DEGs were mainly enriched in three major GO functional groups, including biological processes (BP), molecular functions (MF), and cellular components (CC). In the shoots, 447 out of 1291 specifically up-regulated DEGs were significantly enriched in 67 sub-classes (Appendix A), and the most enriched terms were protein kinase activity, DNA binding, and protein phosphorylation (Appendix A). A total of 1679 out of 3274 specifically down-regulated DEGs were significantly enriched in 190 sub-classes, such as hydrolase activity, hydrolyzing O-glycosyl compounds, apoplast, and carbohydrate metabolic process (Appendix A). In the roots, 1320 out of 2955 specifically up-regulated DEGs were significantly enriched in 110 sub-classes, such as membrane, o-methyltransferase activity, and heme binding (Appendix A). A total of 622 out of 1512 specifically down-regulated DEGs were enriched in 115 significant sub-classes, such as nicotianamine synthase activity, nicotianamine biosynthetic process, and manganese ion binding (Appendix A). It is worth noting that protein phosphorylation and metal ion transport were significantly enriched in both the shoots and roots via specifically up-regulated DEGs, indicating that post-translational modification played an important role in the response to LN stress in wheat. Moreover, 43 out of 81 co-up-regulated DEGs in the roots and shoots were enriched in 35 sub-classes (Figure 4A and Appendix A), including nitrate reductase (NADPH) activity and nitrate assimilation. Further, 26 out of 89 co-down-regulated DEGs in the roots and shoots were enriched in 17 sub-classes (Figure 4B and Appendix A), such as allantoin catabolic process, integral component of membrane, and allantoinase activity. In addition, 138 of the 357 DEGs up-regulated in roots and down-regulated in shoots were enriched in 41 subclasses (Appendix A), including xylan biosynthetic process, cellulose synthase (UDP-forming) activity, and membrane.

KEGG pathways were classified into five categories, including metabolism, genetic information processing, environmental information processing, cellular process, and organismal systems, with the most abundant DEGs identified in pathways belonging to metabolism (Appendix A). In the shoots, 490 out of 1291 specifically up-regulated DEGs were significantly enriched in 15 pathways (Appendix A), with the most significantly enriched being the MAPK signaling pathway (Appendix A). A total of 1485 out of 3274 down-regulated DEGs in the shoots were enriched in 78 significant pathways (Appendix A), with phenylpropanoid biosynthesis, and cutin, suberin, and wax biosynthesis being the most significant (Appendix A). In the roots, among the 57 pathways significantly enriched for the 1030 out of 2955 up-regulated DEGs (Appendix A), phenylpropanoid biosynthesis and flavonoid biosynthesis were the most significant pathways (Appendix A). A total of 322 out of 1512 specifically down-regulated DEGs were significantly enriched in 31 pathways (Appendix A), and the most significantly enriched pathways were cysteine and methionine metabolism, biosynthesis of various plant secondary metabolites, and nitrogen metabolism (Appendix A). There were also 52 pathways enriched in the roots via up-regulated DEGs and in the shoots via down-regulated DEGs. In addition, 17 out of 81 co-up-regulated DEGs in the roots and shoots were enriched in four pathways (Figure 4C and Appendix A), including amino sugar and nucleotide sugar metabolism, starch and sucrose metabolism, and linoleic acid metabolism. Four co-down-regulated DEGs in the roots and shoots were only enriched in nitrogen metabolism (*TraesCS4A01G042000*, *TraesCS7A01G428500*, *TraesCS7B01G328700*, and *TraesCS7D01G420900*) (Figure 4D and Appendix A). In addition, 111 of the 357 DEGs up-regulated in roots and down-regulated in shoots were enriched in 24 pathways (Appendix A), including phenylpropanoid biosynthesis, cutin, suberin, and wax biosynthesis, and flavonoid biosynthesis.

In conclusion, although some genes were involved in functions in both the roots and shoots, there was tissue specificity in the response to LN between the shoots and roots in wheat.

### 2.5. Identification of Differentially Expressed Transcription Factors and Protein Kinases

TFs and PKs play an essential role in the regulation of gene expression, especially in response to biotic and abiotic stresses. In this study, 6291 TFs belonging to 67 families were identified in wheat. A total of 541 TFs were differentially expressed in shoots and/or roots after LN stress (Figure 5A and Appendix A). In the shoots, 241 differentially expressed TFs, including 123 up-regulated and 118 down-regulated DEGs, belonged to 34 transcription factor (TF) gene families. Notably, members of several TF gene families (9/34 families) were all down-regulated in the shoots, such as the Tify (10/10 genes) and B3-ARF (7/7 genes) families, but 13 TF gene families, such as the MYB-related (38/38 genes) and C2C2-CO-like (2/2 genes) families, were all up-regulated in the shoots (Appendix A). In the roots, 333 differentially expressed TFs belonged to 32 TF gene families. Of these 333 TFs, 79 were down-regulated and 254 were up-regulated. In particular, in many TF gene families (13/32), all TFs were up-regulated in the roots, e.g., HB-BELL (9/9 genes) and NF-YA (11/11 genes) (Appendix A). Simultaneously, all TFs in six families were down-regulated in the roots, such as the LOB (12/12 genes) and RWP-RK (3/3 genes) families (Appendix A). In addition, there were 33 differentially expressed TFs that overlapped in the shoots and roots, and these genes belonged to 11 TF gene families (Figure 5B). We also found that 18 of 33 genes were up-regulated in the roots but down-regulated in the shoots, belonging to seven families, such as AP2/ERF-ERF (four genes) and MYB (three genes). Overall, TFs play an important role in the response to LN stress in wheat, and these genes may function in a tissue-specific manner to respond to LN stress.

We also identified 5326 PKs in wheat. A total of 378 PKs were differentially expressed in the shoots and/or roots after LN stress. There were 203 differentially expressed PKs in the shoots, of which 87 were down-regulated and 116 were up-regulated, belonging to 38 families (Figure 5C and Appendix A). Members of many protein kinase (PK) gene families (20/38) were down-regulated, for example, RLK−Pelle_RLCK−V (6/6 genes) and RLK−Pelle_LRR−III (5/5 genes). The family members of RLK-Pelle_LRK10L-2 (17/17 genes) and RLK-Pelle_SD-2b (6/6 genes) were all up-regulated in the shoots. In the roots, 191 differentially expressed PKs were identified, of which 42 were down-regulated and 149 were up-regulated, from 34 PK gene families (Appendix A). Notably, members of some of the PK gene families (25/34 families) were predominantly up-regulated, such as the RLK−Pelle_DLSV (35/39 genes) and RLK−Pelle_WAK (24/28 genes) families. Moreover, 16 differentially expressed PKs overlapped between the roots and shoots (Figure 5D), and these genes belonged to 11 PK gene families (Figure 5B). Nine PKs were expressed with different trends in the roots and shoots. Thus, PKs also play an important role in the development and adaptation to LN stress, and may have tissue-specific roles.

### 2.6. Expression Pattern of Nitrate Signaling Pathway Genes

The genes of the N signaling pathway play an important role in signal transduction in plants under LN stress. In general, homologous genes have similar functions across different species. As a complex hexaploid species, identifying and analyzing the expression patterns of N signaling pathway genes in wheat is important for us to understand the regulation mechanism of wheat N utilization.

A total of 107 N pathway homologous genes were identified in wheat (Appendix A). Overall, 89 of the 107 N signaling pathway homologous genes had detectable transcript levels in wheat roots and/or shoots, while the remaining 18 genes were not expressed (FPKM < 1) (Figure 6). Thirty-two of the forty homologous genes involved in N-signal perception had detectable expression. These genes were homologous to NRT1.1 (10/13), CNGC15 (3/6), CIPK8 (3/3), CIPK23 (3/3), CBL1/9 (4/6), and SnRK2s (9/9) (Figure 6). Among these homologous genes, two SnRK2s homologous genes were up-regulated (*TraesCS4A01G235600* and *TraesCS4B01G079300*) and six NRT1.1 (OsNRT1.1B) homologous genes were down-regulated in the roots under LN stress (Appendix A). Additionally, the expression trends of CIPK8 and CBL1/9 homologous genes were completely opposite between the roots and shoots, suggesting different LN response patterns. For the regulatory hub, NLP and its interacting proteins acted as key regulators of N signaling by activating many N-responsive genes. In our study, a total of 20 homologous genes were identified as being expressed, including homologous genes of NLP6/7 (3/3), TCP20 (3/3), NRG2 (3/3), CPK10/30/32 (9/9), and HBI1 (2/2) (Figure 6). Among these homologous genes, only one CPK10/30/32 (*TraesCS4A01G283400*) homologous gene was up-regulated in the roots (Appendix A). In both the roots and shoots, the expression trends of the NRG2 and HBI1 homologous genes were completely reversed, and the NLP6/7 homologous genes showed a consistent trend towards down-regulation. These ORGs were further divided into two functional groups. Among the genes that responded positively to LN stress, we identified 21 of 26 homologous genes with detectable expression, including OsNhd1 (3/3), TGA1/4 (3/3), ANR1 (6/6), bZIP (9/9), and NAC56 (0/5) (Figure 6). An up-regulation of a bZIP1 homologous gene was observed in the roots (Appendix A). The expression trends of OsNhd1 were consistently up-regulated in both tissues. On the other hand, among the genes that responded negatively to LN stress, we identified 16 of 21 homologous genes that had detectable expression, including BT1/2 (6/6), HRS1/HHOS1 (3/3), SPL9 (1/6), and LBD37/38/39 (6/6) (Figure 6). In the roots, HRS1/HHOS1, LBD37/38/39, and BT1/2 each had three down-regulated homologous genes (Appendix A). In the shoots, there were two down-regulated homologous genes belonging to HRS1/HHOS1 (*TraesCS2B01G135600*) and LBD37/38/39 (*TraesCS2D01G193400*). In particular, the expression trends of BT1/2 homologous genes were completely opposite in the roots and shoots, whereas the expression trends of HRS1/HHOS1 and LBD37/38/39 were consistently down-regulated in both tissues. In conclusion, these genes played a crucial role in response to nitrate signaling, and had different response patterns in wheat roots and shoots.

### 2.7. Identification of Co-Expression Genes in Response to LN Stress

Genes with highly correlated expression patterns are often involved in the same biological process. We collected 36 available transcriptome datasets of wheat under LN stress and performed WGCNA. A total of 17,840 genes with expression in one or more samples were analyzed and classified into 12 color modules (Figure 7A). Among these modules, only the MEturquoise module (11,195 genes) showed a distinctly different expression pattern in the shoots and roots, with almost all genes (98%) being more highly expressed in the roots than in the shoots (Figure 7B). Among the 1889 roots’ DEGs identified in the MEturquoise module, five TFs (*TraesCS4B01G299400*, *TraesCS4B01G299500*, *TraesCS2A01G281200*, *TraesCS4D01G298400*, and *TraesCS2B01G298600*) had the highest connectivity values (kWithin > 2000), indicating that these genes could be potential hub genes in this module (Figure 7C). In addition, 109 roots’ DEGs of the MEturquoise module were directly connected to all potential hub TFs, and they all had a connectivity of >0.4 (Figure 7D). A total of 109 genes were significantly enriched with 23 GO terms, such as methyltransferase activity, response to oxidative stress, and response to stress (Figure 7E). Five KEGG pathways were significantly enriched, such as phenylpropanoid biosynthesis, glutathione metabolism, and tryptophan metabolism (Figure 7F). In summary, we identified five potential key TFs that regulate the LN response, suggesting their important role in response to LN stress in the roots.

## 3. Discussion

### 3.1. LN Stress Response in Wheat

Abiotic stresses, such as LN stress, usually trigger drastic molecular responses during plant growth [59]. In this study, we constructed transcriptomic datasets of the roots and shoots of wheat seedlings after 12 days of LN stress. Overall, 8468 non-redundant DEGs were identified. DEGs were mainly up-regulated in the roots (66.15%, 2955 up/4467 DEGs), but down-regulated in the shoots (71.62%, 3274 down/4565 DEGs) (Figure 3), suggesting different response patterns between wheat roots and shoots. A previous study on wheat revealed a similar mechanism [51]. After 10 days of LN stress on wheat seedlings (Wanmai No. 52) at the two-leaf stage, 74% of root DEGs were up-regulated, while 76% of leaf DEGs were down-regulated [51]. However, after a short period of LN stress, many DEGs were still predominantly up-regulated in wheat roots, but more DEGs were up-regulated than down-regulated in the shoots [52,53,54]. For example, following 24 h of LN stress on seedlings 21 days after germination of wheat, 83% (PBW677 high NUE) and 96% (PBW703 low NUE) of DEGs were up-regulated in the roots, and 74% (PBW677 high NUE) and 78% (PBW703 low NUE) of DEGs were up-regulated in the shoots [52]. During the one-leaf one-heart period, wheat seedlings were subjected to 12 h of LN stress, resulting in 63% (XM26 high NUE) and 60% (LM23 low NUE) of DEGs being up-regulated in the roots, while 67% (XM26 high NUE) and 59% (LM23 low NUE) of DEGs were up-regulated in the shoots [54]. Interestingly, a similar pattern of response has been found in other species. For example, in *Brassica napus* roots, the up-regulated DEGs were maintained more than the down-regulated DEGs with increasing LN stress time [45]. In rice roots stressed with LN for 15 days, 52% (IR 64) and 93% (Nagina 22) of DEGs were up-regulated [47]. A comparable response pattern was observed in shoots under prolonged LN stress. For example, more down-regulated DEGs were maintained in maize leaves after 20 and 30 days of LN stress [49]. In the leaves of watermelon seedlings, 53% of the DEGs were down-regulated after 14 days [44]. Thus, we hypothesize that the timing of LN stress induces different response mechanisms in plant roots and shoots.

### 3.2. Nitrate Signaling Network in Wheat

Many genes in the hexaploid wheat genome were lost during evolution, while many others were greatly expanded [60]. In identifying signaling pathway genes, we found that some genes in wheat had exactly three homologous genes belonging to the A, B, and D subgenomes, including CIPK8, CIPK23, CBL1/9, SnRK2s, OsNhd1, BT1/2, NRG2, TCP20, and CPK10/30/32. In addition, there were a number of genes with significantly increased numbers of homologous genes, including NRT1.1, CNGC15, ANR1, bZIP1, SPL9, and NAC56. A reduced number of homologous genes were also found for some genes, including TGA1/4, HRS/HHO1, LBD37/38/39, NLP6/7, and HBI1. This may indicate that pathway genes with significantly more homologous genes play a more important role in wheat.

The plant nitrate signaling pathway is a complex network involving numerous genes, and its molecular mechanisms have been extensively studied [9,10,11,12,13]. It is broadly divided into three parts: N signaling perception genes, regulatory hub genes, and ORGs [45]. NRT1.1 was the first nitrate transporter protein cloned in *Arabidopsis* in relation to nitrate uptake [14,15]. In rice, its homologous gene, OsNRT1.1B, is thought to be the main gene responsible for the difference in NUE between indica and japonica rice varieties [61]. In addition, OsNRT1.1B is abundantly expressed in rice roots and down-regulated under LN stress [62,63]. In our study, the expression of all OsNRT1.1B homologous genes (6/6) was significantly down-regulated in wheat. The expression of *TraesCS1A01G210900*, *TraesCS1B01G224900*, and *TraesCS1D01G214200* was abundant, then decreased extremely significantly (${log}_{2}$Foldchange < −6). We speculated that these genes might play an important role in wheat and have gene-editing potential. Notably, NLP7 was found to be an important regulator of nitrate signaling [20,21]. Overexpression of NLP7 in *Arabidopsis* significantly improved plant growth under LN conditions, while also regulating other TFs [20,21]. We found no significant differential changes in the three homologous genes of NLP6/7 in wheat under LN stress. In addition, a previous study found that NLP7 expression was significantly up-regulated in the roots and shoots of high NUE wheat [64]. We speculated that in wheat varieties with different NUE potential, there would be different trends in NLP7 expression after LN stress. In addition, there were several TFs in the nitrate signaling pathway that had a negative regulatory effect. Several studies have reported that HRS1/HHO1 in *Arabidopsis* represses the expression of other NUE-related genes, and also plays a role in root development [31,32,33]. Its expression decreased rapidly under LN stress [31,32]. The HRS1 homologous gene in rice, NIGT1, also performs the same function [65]. LBD37/38/39 and BT1/2 acted as negative regulators of nitrate response genes in response to N starvation [34,35,36]. In our study, the expression of all homologous genes of HRS1/HHO1 (3/3) and some homologs of LBD37/38/39 (3/6) and BT1/2 (3/6) were significantly down-regulated in wheat roots under LN stress. We predicted that reduced expression of negative regulators would play an important role in wheat root growth. In conclusion, the nitrate signaling pathway in wheat is an extremely complex process. A better understanding of the wheat nitrate pathway and its associated genes has been achieved, providing genetic resources and new insights for further research into N signaling mechanisms.

## 4. Materials and Methods

### 4.1. Plant Materials and Treatment

The wheat cultivar Chinese Spring was selected as the plant material. Seeds were surface sterilized with 30% sodium hypochlorite for 20 min and then rinsed with sterile water. The seeds were vernalized at 4 °C for 5 days, and then transferred to 25 °C for 7 days of cultivation. The seedlings were then transferred to a hydroponic system at 25 °C and the entire root system was kept submerged in nutrient solution. Two different nutrient solutions were used, NN (16 mM/L) and LN (3.2 mM/L). The nutrient solution was also changed every three days. On day 12 after nutrient treatment, root and shoot tissues from each plant were sampled and stored at −80 °C for subsequent RNA extraction and analysis.

### 4.2. RNA Isolation, Library Construction and Sequencing

Total RNA was isolated from the roots and shoots using the Plant RNA Kit (TIANGEN). The degradation and contamination of the RNA was monitored on 1% agarose gels. RNA purity, concentration, and integrity were checked using a NanoPhotometer^®^ spectrophotometer (IMPLEN, Westlake Village, CA, USA), a Qubit^®^ RNA Assay Kit in Qubit^®^2.0 Flurometer (Life Technologies, Carlsbad, CA, USA), and an RNA Nano 6000 Assay Kit of the Bioanalyzer 2100 system (Agilent Technologies, Santa Clara, CA, USA). Sequencing libraries were generated with the NEBNext^®^ UltraTM RNA Library Prep Kit for Illumina^®^ (NEB, Ipswich, MA, USA), and 150 bp paired-end reads were generated by sequencing the library preparations on an Illumina Hiseq platform.

### 4.3. Read Preprocessing and Identification of DEGs

The quality of reads was assessed using FastQC (http://www.bioinformatics.babraham.ac.uk/projects/fastqc, accessed on 1 March 2022), and low-quality reads were removed using Trimmomatic [66]. The wheat reference genome and annotation files were downloaded from the International Wheat Genome Sequencing Consortium (IWGSC) [67]. Hisat2 was used to align the reads to the reference genome [68]. We used SAMtools to convert sam files to bam files, as well as sorting and indexing them [69]. Transcript assembly and differential analysis were performed using StringTie and DESeq2 [70,71]. Genes with FPKM ≥ 1.0 in at least one sample were considered to be expressed. Genes satisfying *p*-value < 0.05 and |${log}_{2}$Foldchange| ≥ 1 were defined as DEGs. Data were manipulated and visualized using the R packages tydiverse and ggplot2. PCA was performed using the R packages factoextra and FactoMineR. Hierarchical clustering was performed and visualized as a heatmap using the R package pheatmap. Pearson coefficients were calculated using the R package corrplot.

### 4.4. Function Annotation

GO enrichment analysis was used from the functional annotation files downloaded from IWGSC (https://urgi.versailles.inra.fr/download/iwgsc/IWGSC_RefSeq_Annotations/v1.0/iwgsc_refseqv1.0_FunctionalAnnotation_v1.zip, accessed on 15 March 2022) as background files. KEGG pathway enrichment analysis was used with KofamKOALA [72,73,74,75]. The significance of each pathway was calculated using the hypergeometric distribution test and the results were recorded as *p*-value. Pathways with *p*-value of <0.05 were defined as significantly enriched in GO and KEGG. Identification and classification of TFs and PKs were performed using iTAK (http://bioinfo.bti.cornell.edu/cgi-bin/itak/index.cgi, accessed on 10 June 2022) [76].

### 4.5. Identification of Orthologous Genes

To identify the N signaling pathway genes in the wheat genome, the protein sequences of 34 known *Arabidopsis* and/or rice N signaling genes collected from past studies as queries (Appendix A). The other gene family members of the queried genes in *Arabidopsis* (https://www.arabidopsis.org/, accessed on 15 October 2022) and rice (https://rapdb.dna.affrc.go.jp/, accessed on 15 October 2022) were downloaded as flank genes. The Pfam domains of the target gene were searched using InterPro (https://www.ebi.ac.uk/interpro/, accessed on 20 October 2022). Corresponding hidden Markov model (HMM) files were downloaded from the Pfam database [77]. HMMsearch was used to identify wheat proteins containing the corresponding domains, with an E-value threshold of 1E-5. The obtained non-redundant protein sequences were submitted to SMART (https://smart.embl.de/smart, accessed on 30 October 2022) and InterPro (https://www.ebi.ac.uk/interpro/, accessed on 30 October 2022) to determine protein domains [78]. Multiple sequence alignment of the selected genes in wheat and all family genes of the target gene was performed using MAFFT (https://www.ebi.ac.uk/Tools/msa/mafft/, accessed on 30 October 2022) [79]. Subsequently, a phylogenetic tree was constructed in MEGA using the neighbor-joining algorithm. The candidate genes were considered as positive hits based on the following three criteria: (1) the genes were most closely related to known *Arabidopsis* N signaling genes in the phylogenetic tree (NJ tree); (2) the genes shared a higher sequence similarity to that of the known *Arabidopsis* N signaling genes; (3) the queries sequence and wheat sequence were together located in the smallest branch. The DNA, CDS, and protein sequences of candidates were obtained via IWGSC.

### 4.6. Weighted Gene Co-Expression Network Analysis

To detect key co-expression modules and key genes under LN stress, we generated a co-expression network using the WGCNA package in R [80]. Data from other relevant wheat N studies were obtained from NCBI [54,81]. Genes with low coefficients of variation (CV < 1) were removed and the remaining 17,840 genes were used for analysis. The determination of the soft thresholding power was based on the scale-free topology model fit (R²) ≥ 0.8 using the pickSoftThreshold function. Subsequently, the automatic network construction function (blockwiseModules) was used to complete the network construction and module detection to obtain highly correlated modules. TOMType was signed, the minModuleSize was 50, the soft threshold (power) was 24, and the mergeCutHeight was 0.25. Co-expression and transcriptional regulatory networks were plotted using Cytoscape [82].

## 5. Conclusions

In this study, we characterized the molecular mechanisms of wheat responses to LN stress by transcriptome analysis. Our data suggested that the roots and shoots of wheat had different LN response patterns, such as differences in seedling growth phenotypes and the number of DEGs. Moreover, our results suggested that tissue-specific expression of TFs, such as the MYB-related (38 genes), might also be one of the factors contributing to the differential response of wheat roots and shoots to LN stress. Furthermore, the identification of N signaling genes had defined 20 differentially expressed N signaling regulated genes in wheat. In addition, the WGCNA analysis of 47 transcriptome datasets screened five potential hub regulatory LN response genes (*TraesCS4B01G299400*, *TraesCS4B01G299500*, *TraesCS2A01G281200*, *TraesCS4D01G298400*, and *TraesCS2B01G29860*) in wheat. Overall, our results revealed the LN response mechanism in wheat and screened out five potential candidate hub regulatory genes. Our work will provide new insights into the LN stress response mechanism and contribute potential gene expression resources for subsequent genetic breeding in wheat.

## Figures and Tables

**Figure 1 plants-13-00371-f001:**
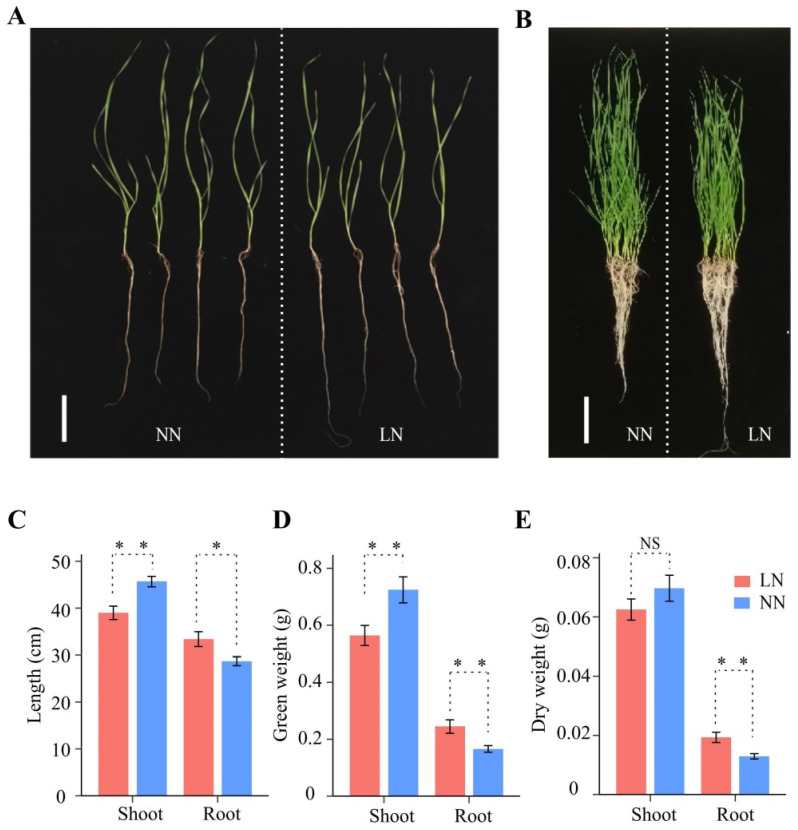
Effects of different nitrogen rates on the wheat crop. (**A**) Four randomly selected seedlings at different nitrogen levels, with NN on the left and LN on the right. Scale bars, 10 cm. (**B**) All seedlings at different N levels, with normal nitrogen (NN) on the left and low nitrogen (LN) on the right. Scale bars, 10 cm. (**C**–**E**) Growth length (**C**), fresh weight (**D**), and dry weight (**E**) of roots and shoots under NN and LN conditions. Bar graphs from left to right represent shoots under LN conditions, shoots under NN conditions, roots under LN conditions, and roots under NN conditions, respectively. Significance levels were estimated using the *t*-test, ** *p* < 0.01, * *p* < 0.05. NS, not significant.

**Figure 2 plants-13-00371-f002:**
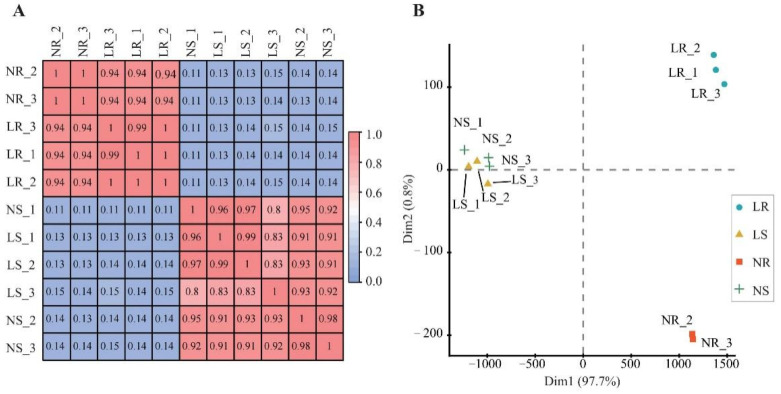
Correlation of wheat samples under different degrees of nitrogen stress. Pearson’s correlation coefficient matrix (**A**) and principal component analysis (**B**) distribution of clustering results. LR, low nitrogen root; NR, normal nitrogen root; LS, low nitrogen shoot; NS, normal nitrogen shoot.

**Figure 3 plants-13-00371-f003:**
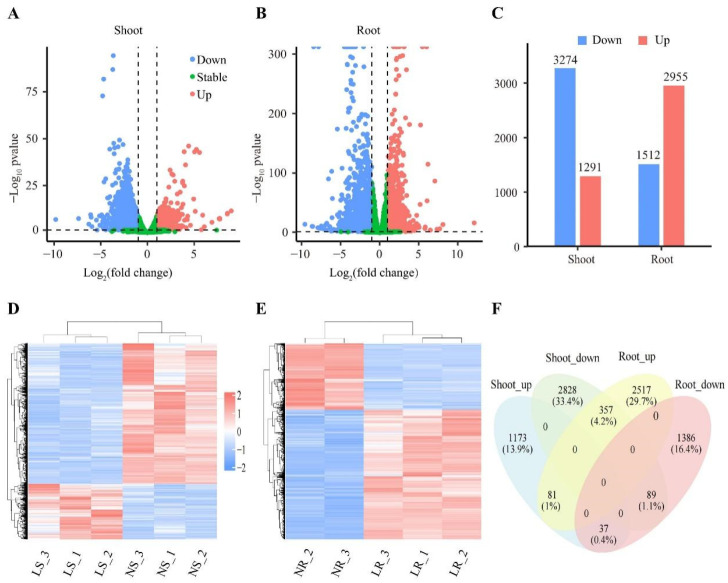
Classification analysis of differentially expressed genes (DEGs). Volcano map of wheat shoots (**A**) and roots (**B**) under LN stress. Each point represents a gene, red points are up-regulated DEGs, blue points are down-regulated DEGs, and green points are complementary differential genes. (**C**) Number of DEGs in wheat roots and shoots. Cluster plots of DEGs in wheat shoots (**D**) and roots (**E**) under LN stress. Colors represent gene expression. High expression is red, and low expression is blue. Venn diagram of up-regulated and down-regulated differentially expressed genes (**F**) in wheat roots and shoots. Genes as a percentage of all genes in brackets. LR, low nitrogen root; NR, normal nitrogen root; LS, low nitrogen shoot; NS, normal nitrogen shoot.

**Figure 4 plants-13-00371-f004:**
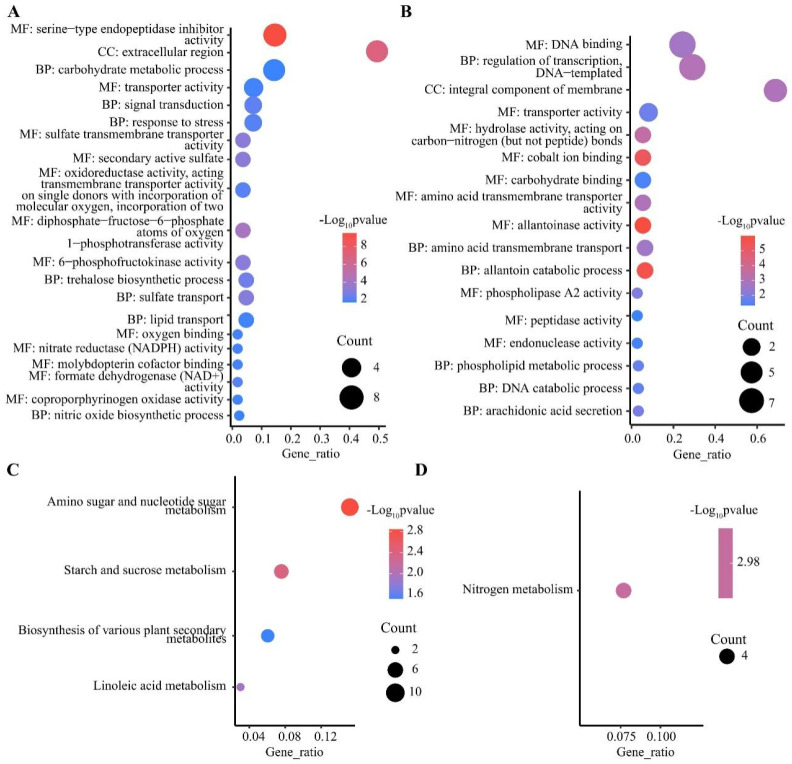
GO and KEGG enrichment analysis of DEGs. GO enrichment analysis of co-up-regulated (**A**) and co-down-regulated (**B**) DEGs in shoots and roots, top 20 pathways by *p*-value. KEGG enrichment analysis of co-up-regulated (**C**) and co-down-regulated (**D**) DEGs in shoots and roots. BP, biological processes; MF, molecular functions; CC, cellular components.

**Figure 5 plants-13-00371-f005:**
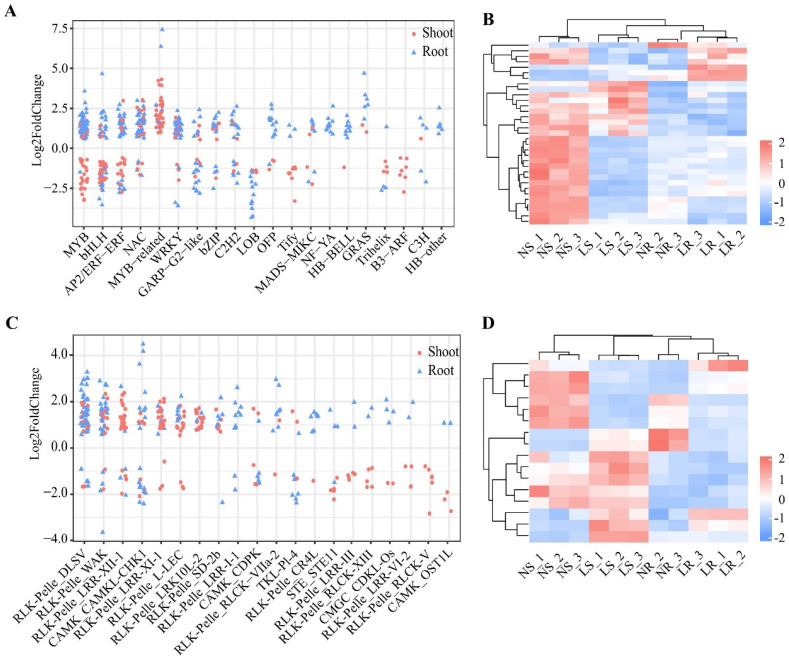
Differentially expressed transcription factors (TFs) and protein kinases (PKs) of wheat under different nitrogen conditions. Classification of differentially expressed TFs (**A**) and PKs (**C**) in roots and shoots, top 20 families by number of genes. Heatmap of co-differentially expressed TFs (**B**) and PKs (**D**) in shoots and roots. LR, low nitrogen root; NR, normal nitrogen root; LS, low nitrogen shoot; NS, normal nitrogen shoot.

**Figure 6 plants-13-00371-f006:**
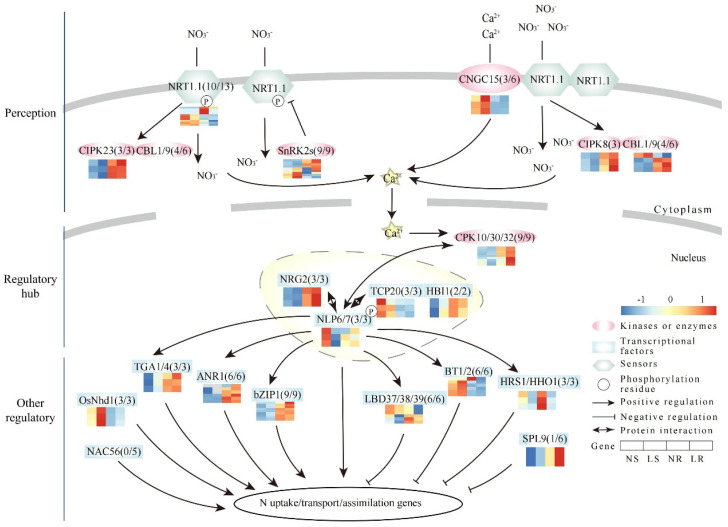
Expression patterns of nitrate signaling pathway genes in wheat. Nitrate is sensed by NRT1.1. The nitrogen signal is transmitted to NLP6/7 via calcium accumulation and phosphorylation. NLP6/7 interacts with several TFs to regulate other regulatory genes downstream, thereby influencing nitrate uptake, translocation, and assimilation. The number following the gene name indicates the number of homologous genes and the number of genes with expression in wheat. ANR, ARABIDOPSIS NITRATE REGULATED; BT, BTB and TAZ DOMAIN PROTEIN; bZIP, BASIC LEUCINE ZIPPER; CBL, CALCINEURIN B-LIKE PROTEIN; CIPK, CBL-INTERACTION PROTEIN KINASE; CNGC, CYCLIC NUCLEOTIDE-GATED CHANNEL PROTEIN; CPK, CALCIUM-SENSOR PROTEIN KINASE; HBI, HOMOLOG OF BRASSINOSTEROID ENHANCED EXPRESSION2 INTERACTING WITH IBH1; HHO, HRS1 HOMOLOG; HRS, HYPERSENSITIVITY TO LOW PI-ELICITED PRIMARY ROOT SHORTENING; LBD, LATERAL BOUNDARY DOMAIN-CONTAINING PROTEIN; N, nitrate; NAC, NAM-ATAF-CCUC DOMAIN-CONTAINING PROTEIN; Nhd1, N-MEDIATED HEADING DATA-1; NLP, NIN-LIKE PROTEIN; NRG, NITRATE REGULATORY GENE; NRT, NITRATE TRANSPORTER; SnRK2, SUCROSE NON-FERMENTING1 (SNF1)-RELATED PROTEIN KINASE 2S; SPL, SQUAMOSA PROMOTER BINDING PROTEIN-LIKE; TCP, TEOSINTE BRANCHED1/CYCLOIDEA/PROLIFERATING CELL FACTOR; TGA, TGACG MOTIF-BINDING FACTOR. LR, low nitrogen root; NR, normal nitrogen root; LS, low nitrogen shoot; NS, normal nitrogen shoot.

**Figure 7 plants-13-00371-f007:**
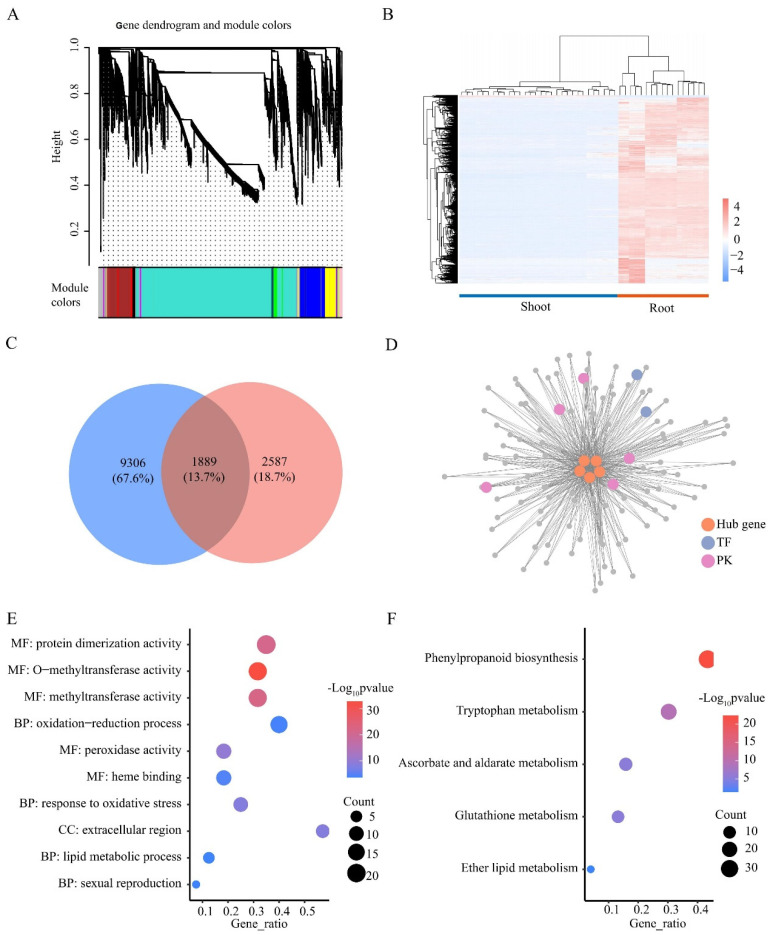
WGCNA analysis of genes expressed in roots and shoots. (**A**) WGCNA compartmentalizes genes into modules. (**B**) Heatmap of FPKM of genes in the MEturquoise module. Colors represent gene expression. High expression is red, and low expression is blue. (**C**) Venn plots of MEturquoise module genes and root DEGs. (**D**) Primary co-expression network of *TraesCS4B01G299400*, *TraesCS4B01G299500*, and *TraesCS2A01G281200*, *TraesCS4D01G298400*, *TraesCS2B01G298600*. GO (**E**) and KEGG (**F**) enrichment analysis of genes in Figure 7D. BP, biological processes; MF, molecular functions; CC, cellular components.

**Table 1 plants-13-00371-t001:** Summary of Illumina sequencing data and mapping results.

Libraries	Row Reads	Clean Reads	GC (%)	Error Rate (%)	Q20 (%)	Q30 (%)	Mapping Alignment Rate (%)	Uniq Mapped Reads (%)	Multi Mapped Reads (%)
LS_1	52,362,666	49,208,442	56.07	0.02	98.06	94.49	98.63	90.87	7.76
LS_2	57,022,090	54,302,414	56.97	0.03	97.83	93.99	98.56	91.27	7.29
LS_3	45,084,032	41,501,168	53.32	0.02	98	94.36	98.34	87.72	10.62
LR_1	46,019,256	44,010,914	54.31	0.03	97.8	93.93	97.85	91.23	6.61
LR_2	43,574,686	41,582,874	54.66	0.03	97.89	94.17	97.75	91.13	6.62
LR_3	43,976,886	41,912,238	53.93	0.03	97.77	93.8	97.74	90.83	6.91
NS_1	56,016,618	53,647,952	58.09	0.03	97.74	93.81	98.61	90.39	8.23
NS_2	46,546,442	44,198,178	55.93	0.03	97.88	94.14	98.61	89.45	9.17
NS_3	49,068,108	46,009,472	55.32	0.02	97.96	94.3	98.43	81.45	16.99
NR_2	49,085,898	46,338,388	53.65	0.02	97.97	94.3	97.24	90.09	7.15
NR_3	47,677,344	44,892,372	53.08	0.03	97.32	92.35	97.18	89.83	7.35

Note: Low nitrogen shoot (LS); Low nitrogen root (LR); Normal nitrogen shoot (NS); Normal nitrogen root (NR).

## Data Availability

All Illumina reads in this study were submitted to the National Center for Biotechnology Information as accession PRJNA997636.

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
