# Peer review of "Transcriptome Profiling Reveals the Gene Network Responding to Low Nitrogen Stress in Wheat"

_plants, 2024, doi:10.3390/plants13030371_

Round 1

Reviewer 1 Report

Comments and Suggestions for Authors

Wang et al, present an interesting finding on how whet root and shoot responds to the low nitrogen stress. Wheat being one of the most important cereal crop, and therefore studying the role of nitrogen deficiency on the growth of wheat would advance the technology for developing better wheat varieties. The authors have provided some great insights into transcriptome profiling under low nitrogen conditions. However, I have few questions for the authors before the manuscript can be accepted for publication:

  1. Line 47-57: In introduction, the authors have cited several studies but have not provided any species reference to which those studies point to. Authors need to mention the respective species. 

  2. Figure 1 Legend is not at all clear for the readers. Especially fig 1A, it is hard to understand what is being shown here. Right and right figure panel in Fig 1A should be differently labelled and described in the legend.  

  3. Line 131-132: Can the authors explain the PCA plots in fig2B and why do they anticipate no effect of nitrogen deficiency on shoots compared to roots ? 

  4. Line 151-153: Linked to the previous comment, the authors show a significant number of genes that are differentially expressed in shoot as well, almost similar to roots. But based on PCA, we do not see such drastic effects of low nitrogen in shoot. Can the authors provide an explanation for this anomaly? 

  5. Fig 3: What was the input used for heatmaps? Is it all genes or only differentially expressed or subset of differentially expressed genes? Same question for PCA too. 

Comments on the Quality of English Language

Some minor proof reading of the Manuscript is required for English. 

Author Response

Reviewer 1

Comments and Suggestions for Authors

Wang et al, present an interesting finding on how whet root and shoot responds to the low nitrogen stress. Wheat being one of the most important cereal crop, and therefore studying the role of nitrogen deficiency on the growth of wheat would advance the technology for developing better wheat varieties. The authors have provided some great insights into transcriptome profiling under low nitrogen conditions. However, I have few questions for the authors before the manuscript can be accepted for publication:

Response: Thank you for your professional suggestions on our manuscript. We have revised the manuscript according to your suggestions, and all changes in the revised manuscript are highlighted in red.

Line 47-57: In introduction, the authors have cited several studies but have not provided any species reference to which those studies point to. Authors need to mention the respective species.

Response: Thank you for your suggestion. Until now, the N signaling pathway have been well established in plants, especially in model plant Arabidopsis. And, the majority of studies on N signaling genes in other species were based on studies in Arabidopsis. Therefore, in our manuscript, we systematically introduced the current research progress in Arabidopsis. As your suggestion, we have revised clear description of genes and species in our revised manuscript.

Figure 1 Legend is not at all clear for the readers. Especially fig 1A, it is hard to understand what is being shown here. Right and right figure panel in Fig 1A should be differently labelled and described in the legend. 

Response: Thank you for your suggestion. We have revised the Figure 1 and the legend to make it easy to understand.

Line 131-132: Can the authors explain the PCA plots in fig2B and why do they anticipate no effect of nitrogen deficiency on shoots compared to roots?

Response: Thank you for your question. In this manuscript, the PCA plots exhibited less variation in shoots than roots under N deficiency. We propose that there are two possible reasons under this phenomenon, including the difference of the organ and their spatial position. Firstly, the function of roots and shoots are different. Roots are the organs that absorbs N, whereas shoots are the organs that utilize and store N. When plants facing N deficiency, shoots can relieve stress by reallocating stored N, whereas the roots lack such a cushioning system. Secondly, roots directly exposure to the N deficiency environment, and we hypothesize that the roots may be more directly response to changes of N condition. Thus, there are more effect on roots compared to shoots under N deficiency.

Line 151-153: Linked to the previous comment, the authors show a significant number of genes that are differentially expressed in shoot as well, almost similar to roots. But based on PCA, we do not see such drastic effects of low nitrogen in shoot. Can the authors provide an explanation for this anomaly?

Response: Thank you for your professional suggestion. In this manuscript, the number of DEGs was similar in roots and leaves, but the PCA results displayed inconsistent results. We suggested that the number of DEGs (Flod change≥2 & p-value<0.05) does not fully reflect the results of PCA. Firstly, PCA was performed using all genes with expression (FPKM≥1 in at least one sample), and any shift in expression would be considered by PCA. This means that PCA was considering all changes in transcript levels, including but not limited to DEGs. Secondly, the DEGs, about 4000 genes, represent only part of the most significant changes in transcript levels relative to the more than 60,000 expressed genes in wheat. Therefore, the contribution of DEGs to PCA variations was limited. In summary, DEGs represent the subset of genes with the most significant changes, but do not represent the changes in transcript levels of all genes (PCA variations). Thanks again for your professional suggestion.

Fig 3: What was the input used for heatmaps? Is it all genes or only differentially expressed or subset of differentially expressed genes? Same question for PCA too.

Response: Thank you for your question. There may be some unclear on the figure legend and the methods, and we have revised it. The data used for heatmaps are the FPKM all differentially expressed genes (Fold change≥2) in shoots (Fig. 3D) and roots (Fig. 3E), respectively. Whereas, the data used for PCA plots are the FPKM of all expressed genes in roots and/or shoots.

Comments on the Quality of English Language

Some minor proof reading of the Manuscript is required for English.

Response: Thank you for your suggestion. We have checked the entire manuscript carefully for language mistakes.

Reviewer 2 Report

Comments and Suggestions for Authors

Dear Authors,

I have reviewed the manuscript and my comments are below:

The abstract currently consists of 339 words - this is much more than what MDPI requires (maximum 250 words). I ask the authors to be happy to correct this. A clearer formulation of the goals is necessary.

Keywords: I don't understand why the authors have to capitalize keywords that are not abbreviations or proper names? I ask them to fix them.

Introduction and References: The literature references used are appropriate, but they must be supplemented in such a way that the literature results published in the last 5 years predominate. This is also requested by MDPI. Thus, the Introductio n chapter will be longer, but that's okay, because it might be short anyway.

The References are not formally appropriate, and I also do not understand what the Figure and table legends section is after the References. Why is this here and what does it represent? Please delete the authors or include them properly in the appropriate text.

Results: Figure signatures should not be above the images, but below them! I ask the authors to fix this.

Intertextual figure and table references in the text for the first time must be followed by the text or figure - this is currently not valid in the text. Please the authors to fix this.

Conclusion chapter is missing, and it would be very necessary. I ask the authors to replace this.

Author Response

Reviewer 2

Comments and Suggestions for Authors

Dear Authors,

I have reviewed the manuscript and my comments are below:

Response: Thank you for your professional suggestions on our manuscript. We have revised the manuscript according to your suggestions, and all changes in the revised manuscript are highlighted in red.

The abstract currently consists of 339 words - this is much more than what MDPI requires (maximum 250 words). I ask the authors to be happy to correct this. A clearer formulation of the goals is necessary.

Response: Thank you for your kindly suggestion. We have shortened the abstract as requested and the word count is now 243 words.

Keywords: I don't understand why the authors have to capitalize keywords that are not abbreviations or proper names? I ask them to fix them.

Response: Thank you for your suggestion. We have revised them in the keywords.

Introduction and References: The literature references used are appropriate, but they must be supplemented in such a way that the literature results published in the last 5 years predominate. This is also requested by MDPI. Thus, the Introduction chapter will be longer, but that's okay, because it might be short anyway.

Response: Thank you for your kindly suggestion. We have replaced some literature published more than a decade ago with literature from the last 5 years, and also added some recent literature in the revised manuscript. Please check it.

The References are not formally appropriate, and I also do not understand what the Figure and table legends section is after the References. Why is this here and what does it represent? Please delete the authors or include them properly in the appropriate text.

Response: Thank you for your suggestion. We have revised the reference format according to the MDPI request. And, the Figure and table legends have been put to the corresponding position in the revised manuscript.

Results: Figure signatures should not be above the images, but below them! I ask the authors to fix this.

Response: Thank you for your suggestion. Figure signatures have been put to the corresponding position.

Intertextual figure and table references in the text for the first time must be followed by the text or figure - this is currently not valid in the text. Please the authors to fix this.

Response: Thank you for your suggestion. The Figure and table have been put to the corresponding position.

Conclusion chapter is missing, and it would be very necessary. I ask the authors to replace this.

Response: Thank you for your suggestion. We have attached a conclusion chapter after the materials and methods.

Round 2

Reviewer 2 Report

Comments and Suggestions for Authors

Dear Authors,

the amendments have been made, I accept the manuscript in this form.